# Survival of Hydrops Fetalis with and without Fetal Intervention

**DOI:** 10.3390/children9040530

**Published:** 2022-04-08

**Authors:** Yu-Yun Huang, Yu-Jun Chang, Lih-Ju Chen, Cheng-Han Lee, Hsiao-Neng Chen, Jia-Yuh Chen, Ming Chen, Chien-Chou Hsiao

**Affiliations:** 1Department of Neonatology, Changhua Christian Children’s Hospital, Changhua 50050, Taiwan; fish9915@gmail.com (Y.-Y.H.); 168708@cch.org.tw (L.-J.C.); 129130@cch.org.tw (C.-H.L.); 19184@cch.org.tw (H.-N.C.); 182288@cch.org.tw (J.-Y.C.); 2Department of Pediatrics, Chung Kang Branch of Cheng Ching Hospital, Taichung 40705, Taiwan; 3Big Data Center, Changhua Christian Hospital, Changhua 50050, Taiwan; 83686@cch.org.tw; 4Department of Post-Baccalaureate Medicine, College of Medicine, National Chung Hsing University, Taichung 40705, Taiwan; 5Department of Genomic Medicine, Changhua Christian Hospital, Changhua 50050, Taiwan; 104060@cch.org.tw; 6Department of Obstetrics and Gynecology, National Taiwan University Hospital, College of Medicine, Taipei 100006, Taiwan; 7Department of Biomedical Science, Dayeh University, Changhua 50050, Taiwan; 8Department of Medical Sciences, National Tsing Hua University, Hsinchu 30013, Taiwan; 9School of Medicine, Kaohsiung Medical University, Kaohsiung 80708, Taiwan

**Keywords:** fetal hydrops, fetal intervention, survival rate, prematurity, intensive care

## Abstract

Objectives: To investigate the survival rate of hydrops fetalis after fetal interventions and neonatal intensive care. Methods: We reviewed the medical records of patients diagnosed with hydrops fetalis from January 2009 to December 2019 at Changhua Christian Children’s Hospital. All cases had abnormal fluid accumulation in at least two body compartments during pre- and postnatal examination. The primary outcome measure was the mortality rate. We also collected information regarding disease etiology, duration of hospital stay, Apgar score, gestational age at birth, initial hydrops fetalis diagnosis, fetal intervention, first albumin and pH levels, and maternal history. Results: Of the 42 cases enrolled, 30 survived and 12 died; the mortality rate was 28.6%. Furthermore, 22 cases received fetal intervention, while 20 cases did not; there was no significant difference in their survival rates (75% and 68%, respectively). Survival rate was associated with gestational age at birth, initial diagnosis time, birthweight, Apgar score, initial albumin and pH levels, and gestational hypertension. Only one case was immune-mediated. Among the nonimmune-mediated cases, the three most common etiologies were lymphatic dysplasia (12/42), idiopathic disorders (10/42), and cardiovascular disorders (5/42). Conclusions: Overall, hydrops fetalis was diagnosed early, and fetal intervention was performed in a timely manner. Preterm births were more frequent, and birthweight was lower in the cases that underwent fetal intervention than in those that did not, but there was no significant between-group difference in mortality. The initial diagnosis time, gestational age at birth, birthweight, Apgar score, and first albumin and pH levels were independently associated with mortality.

## 1. Introduction

Hydrops fetalis is a rare disease with high risks of intrauterine fetal death and prematurity; the survival rate one year after birth is approximately 20–30% [1,2]. Hydrops fetalis is often divided into immune- or nonimmune-mediated etiologies. Over the last two decades, incidences of immune-mediated disease have become much lower than those of nonimmune disease due to the application of general fetoscopic laser coagulation in twin–twin transfusion syndrome, as well as the widespread use of Rh(D) immune globulin (RhIG) to prevent fetal anemia resulting from Rh(D) incompatibility. These have resulted in significant improvements in the survival rate and morbidity of immune-mediated hydrops over time [1,3].

Nonimmune hydrops fetalis, which has more varied symptoms and causes than the immune-mediated type, is targeted according to its presentation via a variety of fetal interventions, including transabdominal thoracocentesis, in utero pleurodesis [4], thoracoamniotic shunt placement for abnormal fluid accumulation in the pleural space [5,6,7], transabdominal tapping for fetal ascites, and intraperitoneal and intra-amniotic injections of drugs for fetal arrhythmia [8] or heart failure [5]. However, unlike immune-mediated hydrops fetalis, only a few studies have compared the survival rate or prognosis of nonimmune hydrops fetalis with and without fetal therapy. While some reported significant differences, others did not [5,6,9,10]. Therefore, the objectives of this study were: (1) to evaluate whether the outcome of hydrops fetalis is associated with fetal intervention, (2) to identify the predictive factors for its survival and prognosis, and (3) to describe the clinical features of liveborn neonates with hydrops fetalis.

## 2. Materials and Methods

This study was performed in the neonatal intensive care unit (NICU) at the Changhua Christian Children’s Hospital, which is a tertiary perinatal and neonatal unit in Central Taiwan. The research related to human use complies with all relevant national regulations and institutional policies, as well as with the tenets of the Helsinki Declaration. The study design was reviewed and approved by the Research Ethics Board Committee of Changhua Christian Children’s Hospital. The requirement for informed consent was waived due to the retrospective nature of this study.

We retrospectively reviewed the medical records of patients admitted to the NICU with a diagnosis of hydrops fetalis from January 2009 to December 2019. We searched for cases with a diagnosis of hydrops fetalis, perinatal intestinal perforation with ascites, pleural effusion, and general edema, inclusive of ICD 10 codes J90, J91.8, P56.0, P56.99, P83.2, P83.3, P83.39, R60.9, R60.1, E84.11, P76.0, P76.1, P76.8, P76.9, P78.0, P78.1, P78.89, and P78.9, as well as ICD 9 codes 773.3, 778.0, 778.5, 511.91, 782.3, 777.1, 777.4, 777.6, 777.8, and 777.9. This study enrolled both nonimmune- and immune-mediated cases. Cases with a medical record of perinatal abnormal fluid accumulation in at least two body compartments were included. We did not capture data on stillbirths, or neonates that expired in the delivery room. Cases with non-available or incomplete medical chat were also excluded. We collected information about the etiology, sex, hospitalization length, Apgar score, gestational age at birth, bodyweight at birth, delivery type, presence of fetal intervention, parvovirus B19 infection status, fetal therapy type, first albumin and pH levels, respiratory condition (ventilator type, inhaled nitric oxide use, and ventilator use duration), duration of inotropic agent use, and maternal history. The first laboratory examinations and imaging for hydrops fetalis after admission to the NICU were established in our order system beginning in 2009. The investigations include red blood cell morphology, Coombs test, reticulocyte count, glucose-6-phosphate dehydrogenase, hemoglobinopathy and thalassemia screening for hematologic disorders, anti-cytomegalovirus immunoglobulin (Ig)M and IgG, a viral culture via ear and rectal swab for identifying possible infectious agents, creatine phosphokinase, creatine kinase–MB mass and troponin-I for heart disease, disseminated intravascular coagulation profile (prothrombin time, activated partial thromboplastin time, fibrinogen, D-dimer) for coagulopathy, arterial gas analysis for evaluation of the condition severity, tandem mass spectrometry (MS/MS) screening for inborn errors of metabolism, and albumin for body fluid distribution evaluation. Echocardiogram, abdominal echogram, and renal echogram were all included in the order package to evaluate congenital organ anomalies. In our study, almost all cases underwent all items of the order package. However, some patients were so critical that therapeutic management had to be done before the examination for diagnosis. If a patient died within hours after admission, the investigation was not completed. Considering the aim of our study, the survival pattern of hydrops fetalis, the patients with an incomplete investigation due to early death were still included for comparison with the survivors. A statistical regression analysis was done for this condition. If the cause of hydrops could not be established based on prenatal or postnatal workup results, it was regarded as having an idiopathic origin.

The mothers of all enrolled patients received antenatal therapy. The attending obstetrician decided the duration and type of antenatal therapy based on personal experience. If a significant amount of fluid in the body cavity was noted before the patient’s birth, the attending obstetrician often performed transabdominal fluid drainage for the fetus’ pleural effusion or ascites followed by immediate delivery. Supportive care, including albumin and blood component transfusion, pigtail insertion for drainage, empiric antibiotic administration, and advanced management based on the patient’s clinical condition, were provided to stabilize the patient’s vital signs after admission. For patients with chylothorax, the treatment was nil per os for >1 week and initial feeding with a medium-chain triglyceride diet. If the symptoms did not resolve, continuous infusion with octreotide was administered. In our study, no case with chylothorax required surgical intervention.

### Statistical Analysis

Statistical comparisons were performed using the Mann–Whitney U test for continuous variables and Fisher’s exact test for categorical variables. Multivariate logistic regression analysis was used to calculate the adjusted odds ratios for mortality. Variables that achieved statistical significance during the univariate analysis were subsequently included in the multivariate analysis. The final model retained only statistically significant factors after multiple regressions. All statistical tests were two-tailed, and *p* < 0.05 was considered statistically significant. The data were analyzed using the IBM SPSS Statistics for Windows, Version 22.0 (IBM Corp., Armonk, NY, USA).

## 3. Results

A total of 42 cases were included. Most cases involved nonimmune hydrops fetalis (41/42), with only one case of immune-mediated disease. The latter was diagnosed with hemolytic anemia due to Rh incompatibility with anti-E + c.

Regarding the etiology of nonimmune-mediated hydrops fetalis, the cases were categorized into the 10 groups suggested by *Avery’s Diseases of the Newborn* [11], with little modification. The most common etiologies of nonimmune-mediated hydrops fetalis were lymphatic dysplasia (12/42), idiopathic disorders (10/42), and cardiovascular disorders (5/42) (Table 1). Previously, Parvovirus B19 infection during pregnancy was considered an important cause of hydrops, but there were no cases of hydrops due to Parvovirus B19 infection in this study. In our institution, a PCR test is used to screen for Parvovirus B19.

Among the 42 cases, 30 cases survived, and 12 cases expired: the mortality rate was 28.6%. Furthermore, 22 cases received fetal intervention, while 20 cases did not. The survival rates of the groups with and without fetal intervention were 68% and 75%, respectively, and this difference was not statistically significant. The gestational week at diagnosis and hemoglobin levels were significantly different between cases who did and did not receive fetal intervention (Table 2). Among the cases that survived after receiving fetal intervention, the gestational age at diagnosis was also significantly earlier than among the survivors who did not receive fetal intervention. The median gestational age at initial diagnosis with and without fetal intervention was 29 and 32 weeks, respectively (*p* = 0.013). Further, the survivors had significantly different birth bodyweights, Apgar scores, initial albumin and pH levels, and rates of gestational hypertension (Table 3). The median lengths of hospitalization in the non-survivor and survivor groups were 1 day and 36.5 days, respectively. Of note, among the cases of survivors, the gestational age at diagnosis was significantly higher in the cases that received fetal intervention. This indicates that prenatal therapy has a close correlation with prolonged intrauterine survival, which allowed the fetuses with hydrops fetalis to reach the appropriate gestational age for birth. A total of 42 cases were enrolled for logistic regression analysis; we observed that the survival rate was related to the gestational age at birth and initial diagnosis time, birth bodyweight, Apgar score, first pH level of gas analysis, and gestational hypertension (Table 4). The factor of maternal gestational diabetes mellitus was also included for analysis, but only one case was diagnosed with gestational diabetes in our study. As seen in Table 5, some patients received two kinds of prenatal intervention. Most cases received intervention for fluid drainage. Only one case that received the intrauterine injection of amiodarone was diagnosed with an atrial flutter prenatally. The case initially presented with ascites, pleural effusion and pericardial effusion at 29 weeks of gestational age. Although the optimal therapy for fetal atrial flutter is controversial, the obstetrician in 2009 considered the patient to be a case of severe hydrops, and amiodarone was a trend for treatment then [12].

## 4. Discussion

According to a recent review study, the most common causes of hydrops fetalis have changed over the past 15 years: similar to our study, lymphatic lesions have become more predominant, and the occurrence of chromosomal abnormalities has declined [1]. Advanced detection of structural anomalies by antenatal sonography [13] and of chromosomal abnormalities by prenatal genetic screening may have increased the rate of prenatal diagnosis of the incurable congenital diseases associated with hydrops fetalis, which results in the early termination of more pregnancies, as well as a change in the major causes of fetal hydrops with liveborn neonates in the NICU over time. The genomic medicine department in our institution provides diverse and effective genetic screening and antenatal counseling about the diagnosis [14], which may explain the absence of cases with chromosomal abnormalities in this study. In addition, cardiovascular malformation, which is often syndromic or associated with chromosomal abnormality, was reported as the most common cause of hydrops fetalis in a large systematic review [1,15]. However, in our research, cardiovascular malformation was not as prevalent as lymphatic dysplasia and idiopathic causes. Another potential explanation may be that a prenatal general obstetrics scan was provided at almost every visit to the obstetrics clinic or hospital department in Taiwan [16]. According to a report by the Ministry of Health and Welfare in Taiwan, over 90% of pregnant women receive a prenatal examination at least once. Termination would be a choice if an uncorrectable congenital disease were found during this examination.

One major limitation of our study was that cases of intrauterine death (either due to a worsening condition or artificial abortion) were not included. This may bias the mortality rate in our study to be lower than the true result. Recent studies that included intrauterine death reported that the survival rate of hydrops fetalis was about 20–30% [17,18], which was lower than the survival rate in our study (71.4%). Differences in diagnoses may have contributed to different survival rates [2,19]. The highest mortality rate was seen among neonatal hydrops cases with a congenital anomaly, and the lowest was seen among neonatal hydrops cases with congenital chylothorax. Hydrops fetalis with a congenital anomaly (either a syndromic or structural anomaly) is often diagnosed in the early period of gestation, which is also associated with high mortality [2]. In cases of congenital chylothorax, pulmonary hypoplasia increases the risk of mortality. If the chylothorax progressed to a syndromic state in the middle or late period of gestation, after lung development, a better prognosis would be expected when compared to other causes. A large review study showed that regardless of the type of prenatal therapy used for fetal hydrothorax, the overall survival rate was 63% [20]. In a recent retrospective study, premature (gestational age < 34 weeks) patients with chylothorax who received a fetal intervention had a significantly higher survival rate than those who did not receive an intervention [6]. However, if the study group were to enroll many types of non-immune hydrops fetalis, as in our study, there would be no significant difference in postnatal mortality between the group with antenatal therapy and that without [10].

In our study, two cases were diagnosed as congenital hypothyroidism through newborn screening. One was diagnosed with hydrops fetalis with ascites, pericardial effusion, and general edema at 37 weeks of gestational age and was delivered immediately one day later. The other patient was diagnosed with hydrops fetalis with pleural effusion, ascites, and pericardial effusion at 30 weeks of gestational age and was delivered via emergency Caesarean section one week later due to fetal cardiac failure and maternal pre-eclampsia with progressive dyspnea. These cases did not receive antenatal intervention and survived after treatment. There was also no maternal history of hypothyroidism. Although congenital hypothyroidism is not usually classified as a cause of hydrops fetalis, one study reported an association [21]. Another study [22] also described a hydrops fetalis case of fetal hypothyroidism with a goiter due to maternal propylthiouracil exposure, which was reversed by intra-amniotic levothyroxine. The symptoms of congenital hypothyroidism rarely develop in the fetus due to its low T4 requirement. The hypothesized mechanism is that deficient adrenergic activity in congenital hypothyroidism might lead to nonimmune hydrops fetalis [22].

Parvovirus B19 infection during pregnancy is a major cause of hydrops fetalis, but our study did not find any cases of parvovirus infection. One investigation in Taiwan reported that the overall prevalence of anti-B19 IgG decreased from 32.8% in the 1980s to 23.1% in the 2000s [23]. Therefore, the absence of parvovirus infection from our study may be related to the lower prevalence of Parvovirus B19 infection in recent decades.

Consistent with our findings, previous studies [5,19] have found that poor prognostic factors for survival include smaller gestational age, a low Apgar score at 5 minutes, lower serum pH level, and a serum albumin concentration lower than 2 g/dL. The serum albumin concentration was considered to be closely correlated with the severity of hydrops fetalis [24]. Anemia or cardiac dysfunction can lead to congestive heart failure with elevated central venous pressure. This mechanism contributes to increased capillary filtration pressure and restriction of lymphatic return. Heart failure contributes to liver congestion, which causes a decrease in albumin synthesis. In addition, hypoxia increases capillary permeability to albumin in asphyxiated hydropic infants. Therefore, hypoalbuminemia causes a decrease in plasma colloid osmotic pressure, enhancing edema formation and serous effusion.

This study had some additional limitations worth noting. First, this was a single center, nonrandomized, retrospective study of neonates that had been admitted to our NICU with a final diagnosis of hydrops fetalis. Second, as previously stated, cases of intrauterine death (either due to a worsening condition or artificial abortion) were not included. Third, the sample size was too small to avoid selection bias. Considering the rarity of hydrops fetalis, it is remarkable that we identified 42 cases within ten years at a single center. Fourth, the decision for antenatal fetal intervention was made at the discretion of the attending obstetrician. Nevertheless, in our study, 95% of fetal interventions (21/22) were performed by the same doctor, which might minimize the risk of bias caused by various indications for fetal intervention. Moreover, the fact that fetal intervention was not performed for mild symptoms of abnormal fluid collection may have resulted in a selection bias [10]. Although the survival rate did not significantly increase after fetal intervention in our study, intervention was closely correlated with prolonged intrauterine survival, which may have allowed the fetus to develop in the late second or early third trimesters and grow to the appropriate gestational age for birth.

## 5. Conclusions

In conclusion, advancement in antenatal sonography, genetic surveys, and interventions have contributed to different etiologies of hydrops fetalis from previous research. Hydrops fetalis cases with fetal intervention were diagnosed at an earlier stage of gestation, had higher rates of preterm births, and had lower birthweights than those without fetal intervention, but mortality did not differ significantly between the groups. The initial diagnosis time, gestational age at birth, birthweight, Apgar score, and initial albumin and pH levels were the independent factors associated with fetal hydrops mortality. Prenatal therapy will prolong survival in utero and allow the hydrops fetus to grow to the appropriate gestational age for birth.

## Figures and Tables

**Table 1 children-09-00530-t001:** Etiology and prognosis.

Etiology and Prognosis	Prognosis		Fetal Intervention	
	Survival (N = 30)	Died (N = 12)	Yes (N = 22)	No (N = 20)
Idiopathic disorder	5	5	4	6
**Cardiovascular disorder**				
Cardiac failure	2		1	1
Prominent Eustachian valve		1	1	
Interatrial aneurysm	1			1
Ectopic atrial rhythm	1		1	
**Renal disorder**				
Congenital nephrotic syndrome		1	1	
**Maternal or placenta condition**				
Placenta chorioangioma		1		1
**Vascular accidents**				
Twin–twin transfusion	1			1
**Nervous system lesions**				
Vein of Galen malformation	1			1
**Hemolytic anemias**				
Rh incompatibility with anti-E + c		1		1
Maternal-fetal transfusion		1		1
**Infections**				
Congenital hepatitis	1			1
**Gastrointestinal conditions**				
Meconium peritonitis	4		3	1
**Lymphatic abnormalities**				
Congenital chylothorax	4	1	3	2
Congenital lymphatic dysplasia	7		6	1
**Pulmonary conditions**				
Cystic adenomatoid malformation of the lung	1	1	2	0
**Congenital hypothyroidism**	2			2

**Table 2 children-09-00530-t002:** With/without fetal intervention.

	Without Fetal Intervention		With Fetal Intervention		
	N	Median	N	Median	*p*-Value
Length of stay (days)	20	12.5 (1–91)	22	35 (1–120)	0.293
Gestational age (weeks)	20	32.5 (25–38)	22	33.0 (24–36)	0.469
GA ^[a]^ at diagnosis (weeks)	16	30.5 (20–38)	21	28.0 (20–33)	0.025
Birth body weight (gm)	20	2481.0 (840–3670)	22	2152.0 (690–3598)	0.257
Apgar score at 1 min	20	6.5 (1–9)	22	5.0 (1–9)	0.268
Apgar score at 5 min	20	8.0 (1–10)	22	7.0 (2–9)	0.580
Albumin (g/dL) Hb (g/dL)	20 18	2.1 (1.0–3.2) 13.3 (1.0–18.5)	22 21	1.7 (1–3) 15.1 (2.1–19.7)	0.283 0.015

*p*-value by Mann–Whitney U Test, ^[a]^ GA: gestational age.

**Table 3 children-09-00530-t003:** Alive/dead cases among hydrops fetalis cases.

	Survivors		Death		
	N	Median	N	Median	*p*-Value
Length of stay (days)	30	36.5 (4–120)	12	1.0 (1–46)	<0.001
Gestational age (weeks)	30	33.0 (28–38)	12	29.0 (24–35)	0.002
Birth body weight (gm)	30	2499.0 (1510–3670)	12	1738.0 (690–3008)	0.006
Apgar score at 1 min	30	6.5 (1–9)	12	2.0 (1–8)	<0.001
Apgar score at 5 min	30	8.0 (5–10)	12	5.0 (1–8)	<0.001
GA ^[a]^ at diagnosis (weeks)	26	30.5 (20–38)	11	27.0 (20–32)	0.022
albumin (g/dL)	30	2.2 (1.3–3.2)	12	1.1 (1–1.5)	<0.001
1st ph.	29	7.3 (6.9–7.4)	10	6.9 (6.9–7.4)	0.001

*p*-value by Mann–Whitney U Test, ^[a]^ GA: gestational age.

**Table 4 children-09-00530-t004:** Independent risk factors of mortality among fetuses with hydrops fetalis.

Parameter	Univariate Analysis (Crude)				Multiple Analysis (Adjusted)			
	Odds Ratio	95% CI		*p*-Value	Odds Ratio	95% CI		*p*-Value
Gestational age	0.650	0.486	0.869	0.004				
Birth body weight	0.998	0.997	0.999	0.006				
A/S (1) ^[a]^	0.474	0.306	0.734	0.001	0.513	0.282	0.934	0.029
A/S (5) ^[b]^	0.407	0.230	0.722	0.002				
GA ^[c]^ at diagnosis	0.859	0.740	0.998	0.047				
Albumin (g/dL)	0.000	0.000	0.129	0.017				
1st pH	0.000	0.000	0.015	0.001	0.000	0.000	0.238	0.026
PIH ^[d]^ (Yes vs. No)	7.000	1.078	45.437	0.041	97.655	1.159	8227.646	0.043

^[a]^ A/S (1): Apgar Score at 1 min, ^[b]^ A/S (5): Apgar Score at 5 min, ^[c]^ GA: gestational age, ^[d]^ PIH: Pregnancy-induced hypertension.

**Table 5 children-09-00530-t005:** Types of antenatal interventions.

Antenatal Intervention	Number
basket shunt	2
pleurodesis + basket shunt	1
thoracentesis + pleurodesis	2
thoracentesis + ascites tapping	2
ascites tapping	6
thoracentesis	8
intrauterine injection of amiodarone	1

## Data Availability

Data available in a publicly accessible repository.

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
