# Peer review of "Survival of Hydrops Fetalis with and without Fetal Intervention"

_children, 2022, doi:10.3390/children9040530_

Round 1

Reviewer 1 Report

The exclusion criteria are presented but we consider that they can be presented much more clearly.

Author Response

Point 1: The exclusion criteria are presented but we consider that they can be presented much more clearly.

 Response 1: We fully agree with the reviewer’s viewpoints. We add the “We did not capture data on stillbirths, neonates that expired in the delivery room. Cases with non-available or incomplete medical chat were also excluded.” in the materials and methods section. (page 2, line 78)

Reviewer 2 Report

Thank you for allowing me to review the following manuscript titled "Survival of hydrops fetalis with and without fetal intervention" by Huang et al

The study aims at investigating the survival rate of hydrops fetalis after fetal interventions in Taiwan using a retrospective single institute study model. The authors report 28% mortality of the 42 cases enrolled. The main finding is that there was no difference in mortality between those who received and those who did not receive fetal intervention. There were no surprises in the factors that were associated with mortality eg: GA, birth wt APGAR etc.

It is well written study. My comments:

Intrauterine injection of amiodarone - What was the indication. It would be useful that in Table 5, the indications are included as well

Maternal environment plays a significant role in the survival. Did the authors consider them or why exclude them in the model when studying mortality eg: Maternal diabetes

Why would the authors only take non survivors for studying regression. I would recommend changing the name of the table 4 from Logistic regression results of non-survivors to "Independent risk factors of mortality among fetuses with hydrops fetalis" and include all patients

Author Response

Point 1: Intrauterine injection of amiodarone - What was the indication. It would be useful that in Table 5, the indications are included as well

Response 1: We thank the reviewer for the positive feedback. The Result has been revised for adding the indication for intrauterine injection of amiodarone. “Most cases received intervention for fluid drainage. The only one case who received the intrauterine injection of amiodarone was diagnosed with atrial flutter prenatally. The initial presentation of the case were ascites, pleural effusion and pericardial effusion at 29 weeks of gestational age. Although the optimal therapy of fetal atrial flutter is controversial, the obstetrician in 2009 considered that the patient was a severe hydrops, and amiodarone was a trend for treatment then.”(page 9, line 156-162). The Reference has been adding the new article. (page 8, line 313-314)

Point 2: Maternal environment plays a significant role in the survival. Did the authors consider them or why exclude them in the model when studying mortality eg: Maternal diabetes

Response 2: In response to the reviewer’s suggestion, the Result has been revised for explanation. “The factor of maternal gestational diabetes mellitus was also included for analysis, but the only one case who was diagnosed with gestational diabetes in our study.” (page 4, line 154-155)

Point 3: Why would the authors only take non survivors for studying regression. I would recommend changing the name of the table 4 from Logistic regression results of non-survivors to "Independent risk factors of mortality among fetuses with hydrops fetalis" and include all patients

Response 3: We apologize for the misleading.  We clarify on peage 5, line 153: Totally 42 cases were enrolled for logistic regression analysis. We thank the reviewer for this excellent suggestion. We have taken the suggestion made by the reviewer and have made following changes. “Table 4. Independent risk factors of mortality among fetuses with hydrops fetalis ”(page 5, line 168)

Round 2

Reviewer 2 Report

the authors have responded appropriately for my comments